# Effects of Dapagliflozin on Volume Status When Added to Renin–Angiotensin System Inhibitors

**DOI:** 10.3390/jcm8060779

**Published:** 2019-05-31

**Authors:** Mie K. Eickhoff, Claire C. J. Dekkers, Bart J. Kramers, Gozewijn Dirk Laverman, Marie Frimodt-Møller, Niklas Rye Jørgensen, Jens Faber, A. H. Jan Danser, Ron T. Gansevoort, Peter Rossing, Frederik Persson, Hiddo J. L. Heerspink

**Affiliations:** 1Complications Research, Steno Diabetes Center Copenhagen, 2820 Gentofte, Denmark; mie.klessen.eickhoff@regionh.dk (M.K.E.); marie.frimodt-moeller@regionh.dk (M.F.-M.); peter.rossing@regionh.dk (P.R.); frederik.persson.01@regionh.dk (F.P.); 2Department of Clinical Pharmacy and Pharmacology, University Medical Center Groningen, 9713 GZ Groningen, The Netherlands; c.c.j.dekkers@umcg.nl; 3Department of Nephrology, University Medical Center Groningen, 9713 GZ Groningen, The Netherlands; b.j.kramers@umcg.nl (B.J.K.); r.t.gansevoort@umcg.nl (R.T.G.); 4Department of internal medicine, Ziekenhuisgroep Twente, 7600 SZ Almelo, The Netherlands; g.laverman@zgt.nl; 5Department of Clinical Biochemistry, Rigshospitalet, 2100 Copenhagen Ø, Denmark; niklas.rye.joergensen@regionh.dk; 6Department of Endocrinology, Herlev & Gentofte Hospital, 2730 Herlev, Denmark; Jens.Faber@regionh.dk; 7Department of Internal Medicine, Erasmus Medical Center, 3015 CN Rotterdam, The Netherlands; a.danser@erasmusmc.nl; 8Faculty of Health and Medical Sciences, University of Copenhagen, 2200 Copenhagen N, Denmark

**Keywords:** SGLT2 inhibitor, dapagliflozin, diabetic nephropathy, heart failure

## Abstract

Sodium glucose co-transporter 2 (SGLT2) inhibitors reduce the risk of heart and kidney failure in patients with type 2 diabetes, possibly due to diuretic effects. Previous non-placebo-controlled studies with SGLT2 inhibitors observed changes in volume markers in healthy individuals and in patients with type 2 diabetes with preserved kidney function. It is unclear whether patients with type 2 diabetes and signs of kidney damage show similar changes. Therefore, a post hoc analysis was performed on two randomized controlled trials (*n* = 69), assessing effects of dapagliflozin 10 mg/day when added to renin–angiotensin system inhibition in patients with type 2 diabetes and urinary albumin-to-creatinine ratio ≥30 mg/g. Blood and 24-h urine was collected at the start and the end of treatment periods lasting six and 12 weeks. Effects of dapagliflozin compared to placebo on various markers of volume status were determined. Fractional lithium excretion, a marker of proximal tubular sodium reabsorption, was assessed in 33 patients. Dapagliflozin increased urinary glucose excretion by 217.2 mmol/24 h (95% confidence interval (CI): from 155.7 to 278.7, *p* < 0.01) and urinary osmolality by 60.4 mOsmol/kg (from 30.0 to 90.9, *p* < 0.01), compared to placebo. Fractional lithium excretion increased by 19.6% (from 6.7 to 34.2; *p* < 0.01), suggesting inhibition of sodium reabsorption in the proximal tubule. Renin and copeptin increased by 46.9% (from 21.6 to 77.4, *p* < 0.01) and 33.0% (from 23.9 to 42.7, *p* < 0.01), respectively. Free water clearance (FWC) decreased by −885.3 mL/24 h (from −1156.2 to −614.3, *p* < 0.01). These changes in markers of volume status suggest that dapagliflozin exerts both osmotic and natriuretic diuretic effects in patients with type 2 diabetes and kidney damage, as reflected by increased urinary osmolality and fractional lithium excretion. As a result, compensating mechanisms are activated to retain sodium and water.

## 1. Introduction

Sodium glucose co-transporter 2 (SGLT2) inhibitors reduce the incidence of heart failure and renal events in type 2 diabetes patients at risk for cardiovascular disease, as well as in patients with diabetes and chronic kidney disease [1,2,3,4]. The early separation of the event curves between the intervention and control group and the only modest reduction in hemoglobin A_1c_ (HbA_1c_) suggest that the long-term benefits conferred by SGLT2 inhibitors are unrelated to improvements in glycemic control [1,2,4]. 

SGLT2 transporters are responsible for glucose and sodium reabsorption in the proximal tubule. Inhibition of SGLT2 promotes the urinary excretion of glucose and sodium, leading to osmotic diuresis and natriuresis. A few studies reported acute increases in urinary volume and sodium levels, supporting the natriuretic/diuretic properties of this drug class [5,6]. These effects dissipated during prolonged treatment, reflecting a transient natriuretic effect with a subsequent new steady state [5,7]. Other studies reported a decrease in plasma volume and interstitial fluid volume during SGLT2 inhibition, which would be in line with their natriuretic/diuretic profile [8,9]. These effects may, at least in part, explain the observed risk reduction of heart failure events in patients with diabetes [1,2,3]. 

Achieving and controlling optimal volume status can be a challenge in patients with diabetic kidney disease, due to the impaired net excretion of sodium. Guideline-recommended treatment for these patients consists of optimizing glucose and blood pressure control, the latter preferably with agents that intervene in the renin–angiotensin–aldosterone system. However, novel drugs that target HbA_1c_ and, at the same time, optimize volume control are a welcome addition to the therapeutic armamentarium for these patients.

Previous studies that assessed the effects of SGLT2 inhibitors on volume markers were performed in either healthy subjects or in patients with type 2 diabetes with preserved renal function, but not in patients with impaired renal function [5,6,7]. In addition, the previous studies did not control for placebo effects and included small populations, which limits the precision of the reported effect sizes and preclude subgroup analyses.

Therefore, we examined the effects of six to 12 weeks of treatment with SGLT2 inhibitor dapagliflozin compared to placebo on markers of volume status in type 2 diabetes patients with albuminuric kidney disease. In addition, we aimed to characterize the effects of dapagliflozin on specific markers of volume status in relevant subgroups.

## 2. Experimental Section

### 2.1. Design and Participants

This was a post hoc combined analysis of two similar studies: the IMPROVE study and the DapKid study. Both were prospective, double-blinded, placebo-controlled cross-over clinical trials designed to assess the albuminuria-lowering effects of dapagliflozin 10 mg/day. The study designs and primary outcomes of both clinical trials were published previously [10,11]. In short, the IMPROVE study enrolled 34 patients with type 2 diabetes, an HbA_1c_ level between 55 and 100 mmoL/moL, and a first morning void urinary albumin creatinine ratio (UACR) ≥100 mg/g and <3500 mg/g from the Department of Internal Medicine Ziekenhuisgroep Twente, the Netherlands. Thirty-three patients completed the study and were included in the primary analysis. In the DapKid study, 36 patients completed the study. These were patients with type 2 diabetes, an HbA_1c_ level >58 mmoL/moL, and a first morning void UACR ≥30 mg/g from Steno Diabetes Center Copenhagen, Denmark. Both studies included patients between 18 and 75 years old with an estimated glomerular filtration rate (eGFR) ≥45 mL/min/1.73 m^2^. All patients were on a stable dose of renin-angiotensin-aldosterone system (RAAS) blocking treatment for at least four weeks prior to randomization. 

Eligible patients were randomly assigned to two successive treatment periods in which they received dapagliflozin 10 mg/day (AstraZeneca, Södertälje, Sweden) or matching placebo added to standard treatment (Appendix A). The IMPROVE study consisted of six-week treatment periods with a wash-out period of six weeks in between. In the DapKid study, patients were assigned to two consecutive 12-week treatment periods without a wash-out period in between. For both studies, the study medication was provided by AstraZeneca, Södertälje, Sweden. The two studies were similar in design. The only difference between the studies was the duration of study periods (six vs. 12 weeks). We merged both studies under the assumption that a new steady state of body-fluid homeostasis arises early after start of SGLT2 inhibition (i.e., day two) [5,7].

Both studies were approved by the regional medical ethics committees (IMPROVE: METC 2014/111; DapKid: H-15006370). All patients gave written consent before any study-related procedures commenced. Both studies complied with the Declaration of Helsinki and Good Clinical Practice Guidelines. The IMPROVE study was registered with the Netherlands Trial Register (NTR 4439) and the DapKid study was registered with ClinicalTrials.gov (identifier NCT02914691). 

### 2.2. Measurements 

Office blood pressure was measured at the beginning and at the end of each treatment period. In both studies, the average of multiple blood pressure readings was recorded. Blood samples, three consecutive first-morning void urine samples, and 24-h urine samples were obtained at the start and end of the two treatment periods. In these blood samples, routine biochemistry assessments were performed, using the Roche COBAS 6000 analyzer series (Basel, Switzerland) in the IMPROVE study and the Ortho Clinical’s VITROS 5600 (Raritan, Somerset, NJ, USA) in the DapKid study. Glomerular filtration rate was estimated (eGFR) using the Chronic Kidney Disease Epidemiology Collaboration (CKD-EPI) equation. In addition, urine and plasma samples were centrifuged, aliquoted, and stored at −80 °C for later assessment of the volume markers: copeptin, renin, N-terminal pro b-type natriuretic peptide (NT-proBNP), and osmolality in the blood. Copeptin was measured as a surrogate marker of vasopressin. Unlike vasopressin, copeptin is easy to measure and is very stable ex vivo [12,13]. Glucose, sodium, and osmolality were also measured in the 24-h urine samples to calculate the fractional urinary sodium excretion, the urinary glucose excretion, and free water clearance. Free water clearance, the excreted solute-free water per unit time, was calculated as urine volume minus osmolar clearance. Osmolar clearance was calculated as ((urine osmolality × urine volume)/plasma osmolality) [8]. In the IMPROVE study, serum- and 24-h urine lithium were measured to calculate the fractional lithium excretion, which is a proxy for proximal tubular sodium reabsorption [14]. Lithium analyses were performed using inductively coupled plasma mass spectrometry (ICP-MS). Automated immunofluorescence assay on a KRYPTOR platform (Thermo Fisher, Waltham, MA, USA) was used to analyze copeptin in the DapKid study and KRYPTOR Compact analyzer (Brahms GMBH, Hennigsdorf, Germany) was used to analyze copeptin in the IMPROVE study. Renin concentration was measured using an immunoradiometric assay (Cisbio, Codolet, France) [15].

### 2.3. Statistical Analysis

Changes in individual outcomes during the intervention period and effects of the treatment were modeled by linear mixed-effects models with a patient-specific random intercept to account for the correlation of repeated measurements within patients. We included sequence, site, treatment, and site-by-treatment interaction as fixed variables in the model. Volume markers measured at the start of the first treatment period were characterized as baseline value. Due to differences in study design, we compared the end of the treatment periods and included the baseline value in the mixed model. Changes in all markers were analyzed in the merged database, as well as in both studies separately. Markers that were not normally distributed were log-transformed before entering the data into the mixed-effects model. The mean percentage changes of these log-transformed volume markers during dapagliflozin therapy versus placebo were derived by (1 – exp[mean change]) x ^−^ 100 and the same was done for the 95% confidence limits. We tested for carry-over effect which was not present in either of the studies.

Secondly, the effects of dapagliflozin on specific volume markers, namely urinary osmolality, NT-proBNP, copeptin, and renin were assessed in the following baseline subgroups: HbA_1c_ level <63 mmoL/moL vs. HbA_1c_ ≥63 mmoL/moL, eGFR <82 mL/min/1.73 m^2^ vs. ≥82 mL/min/1.73 m^2^, UACR <199.7 mg/g vs. ≥199.7 mg/g, and use of diuretics at baseline (yes/no). The median was used to stratify the subgroups. Subgroup analyses were performed by adding relevant subgroups and an interaction term between treatment assignment and subgroup to the mixed models. Data processing and analyses were performed using SAS software, version 9.4 of the SAS System for Windows (SAS Institute Inc., Cary, NC, USA). 

## 3. Results

### 3.1. Baseline Characteristics

Baseline characteristics of the IMPROVE study and the DapKid study were similar, except for renin, glucose, and HbA_1c_ levels, as well as urinary glucose excretion (Appendix A). The mean age of the subjects in the merged database was 62.7 years (8.8). Most of the subjects were male (82.6%), and 44 subjects (63.8%) used diuretics at baseline. The baseline variables related to glucose status, renal function, and volume status can be found in Table 1.

### 3.2. Changes in HbA_1c_, Renal Function, and Markers of Volume Status

Dapagliflozin, compared to placebo, decreased HbA_1c_ by 5.2 mmol/mol (95% confidence interval (CI): from 3.2 to 7.2 mmoL/moL, *p* < 0.01) (Table 1). Estimated GFR was decreased by 4.1 mL/min/1.73 m^2^ (from 2.4 to 5.9 mL/min/1.73 m^2^, *p* < 0.01), and 24-h urine albumin excretion was reduced by 52.0% (from 34.0 to 72.3%, *p* < 0.01), relative to placebo.

Dapagliflozin increased urinary glucose excretion by 217.2 mmol/24 h (from 155.7 to 278.7 mmoL/24 h, *p* < 0.01) and urinary osmolality by 60.4 mOsmoL/kg (from 30.0 to 90.9 mOsmoL/kg, *p* < 0.01), relative to placebo (Table 1 and Figure 1). Fractional sodium excretion was increased by 104.2% (from 19.0 to 189.4, *p* = 0.02), but there was no change in 24-h urinary sodium excretion (Table 1 and Figure 1). There was a 19.6% (from 6.7 to 34.2%, *p* < 0.01) increase in fractional lithium excretion relative to placebo, suggesting that, during chronic treatment with dapagliflozin, sodium reabsorption in the proximal tubule is inhibited (Table 1 and Figure 1). Compared to placebo, dapagliflozin reduced systolic blood pressure by 5.7 mmHg (from 2.3 to 9.1 mmHg, *p* < 0.01), decreased body weight by 1.3 kg, and increased serum sodium and urea, but did not change NT-proBNP (Table 1). Furthermore, compared to placebo, renin increased by 46.9% (from 21.6 to 77.4%, *p* < 0.01) and copeptin increased by 33.1% (from 23.9 to 42.7%, *p* < 0.01; Table 1 and Figure 2). Free water clearance decreased by −885.3 mL/24 h (from −1156.2 to −614.3 mL/24 h, *p* < 0.01), relative to placebo (Table 1 and Figure 1). In general, the changes in volume markers were consistent between both studies (Appendix A).

As shown in Table 2, the effects of dapagliflozin on urinary osmolality, NT-proBNP, copeptin, and renin were consistent in subgroups defined by baseline HbA_1c_, eGFR, albuminuria, and diuretic use. 

## 4. Discussion

The present study examined effects of dapagliflozin on volume markers in type 2 diabetes patients with micro- or macroalbuminuria. Dapagliflozin increased urinary osmolality and fractional lithium excretion, and it decreased blood pressure and weight after 6–12 weeks, compared to placebo. The observed increases in renin and copeptin suggest activation of compensatory mechanisms to retain sodium and water and restore volume homeostasis. Taken together, these data provide more insight into the natriuretic and diuretic effects of SGLT2 inhibition. 

What does this study add to the existing literature? The present data confirm previous studies on the diuretic and natriuretic effects of SGLT2 inhibitors and extend these to a placebo-controlled setting. We also report for the first time the effect of chronic SGLT2 inhibition on fractional lithium excretion as a proxy for tubular sodium reabsorption inhibition, and we demonstrate activation of compensatory mechanisms to restore extracellular volume homeostasis during prolonged SGLT2 inhibition. Finally, prior studies included patients with type 2 diabetes and preserved kidney function, while our studies enrolled patients with type 2 diabetes and kidney damage in whom sodium and fluid homeostasis is often impaired.

Our results indicate that the diuretic effect of dapagliflozin can be attributed to both osmotic diuresis and natriuretic diuresis (Figure 3). The increase in urinary glucose excretion supports an osmotic diuretic effect, while increases in fractional lithium excretion suggest changes in sodium handling during dapagliflozin treatment. We note that the natriuretic effects of SGLT2 inhibitors appear to be weaker compared to traditional diuretics, as evidenced by smaller increases in renin and aldosterone in head-to-head studies [6,7,9]. The osmotic diuresis, thus, seems to be the driving component of the diuretic effects of SGLT2 inhibitors. We also note that the current observations were performed after 6–12 weeks of dapagliflozin treatment when patients established a new steady state in extracellular volume and when sodium intake matched sodium excretion. Hence, it is unlikely that the observed increase in fractional sodium excretion is a product of increased dietary sodium intake, since we did not observe changes in 24-h urinary sodium excretion. Fractional sodium excretion is calculated as the clearance of sodium divided by the creatinine clearance. The increase in fractional sodium excretion might, thus, be a consequence of decreased glomerular filtration.

Inhibition of sodium reabsorption in the proximal tubule and increased sodium delivery in the distal tubule will generally trigger compensatory mechanisms to maintain sodium and fluid homeostasis (Figure 3). Such effects were also observed in our study, even after 6–12 weeks of treatment with dapagliflozin. Firstly, increased sodium delivery in the distal tubule promotes renin secretion by the juxtaglomerular cells [16]. Increased secretion of copeptin, a surrogate marker of vasopressin, is another mechanism activated in the setting of increased diuresis and reduction in body-fluid volumes to promote water retention. The decrease in FWC may be a consequence of increased vasopressin secretion secondary to volume contraction and/or increased glucosuria. Finally, although not measured in our study, a reduction in plasma volume due to diuretic effects decreases medullary blood flow, which is expected to increase the abstraction of water in the descending limb of Henle (which is impermeable to sodium). This in turn leads to an increased sodium concentration, and an increased passive reabsorption of sodium further down the ascending loop of Henle to maintain fluid homeostasis [17]. 

The increases in osmotic and natriuretic diuresis, along with reductions in body weight and systolic blood pressure and the activation of compensatory mechanisms, as described above, indicate that dapagliflozin reduces plasma volume. Plasma volume contraction may also explain the modest, yet statistically significant increases in plasma sodium and urea. The finding that NT-proBNP did not change during dapagliflozin treatment was surprising, but was also observed in other dapagliflozin studies [9]. A possible explanation is that, in most studies, subjects were not evidently volume overloaded and had NT-proBNP levels in the normal range. Future studies in subjects with elevated NT-proBNP levels and volume overload are required to study this in more detail. 

The effects of dapagliflozin on urinary osmolality, NT-proBNP, copeptin, and renin in the overall study population were consistent irrespective of HbA_1c_, eGFR, albuminuria, or diuretic use at baseline. This finding adds to a growing body of evidence that the effects of SGLT2 inhibitors on volume states are consistent across a wide range of patients with type 2 diabetes, regardless of whether subjects used cardiovascular medication (such as diuretics), or whether they had renal impairment.

This study has limitations. Unfortunately, direct measurements of plasma volume were not available for the included studies. However, prior studies already showed that SGLT2 inhibitors decrease extracellular fluid/plasma volume [8,9,18,19]. We recognize that we combined two studies with different durations of treatment periods. The IMPROVE study consisted of six-week treatment periods, while the DapKid study had treatment periods of 12 weeks. Overall, there was generally a consistent trend in the changes of volume markers between both studies, as illustrated in Appendix A. These data support the notion that differences in treatment periods did not influence the observed changes in volume markers, because a new steady sodium and water balance arises after a few days of treatment. There were also differences in baseline glycemic control between patients enrolled in the IMPROVE study (Almelo, Ziekenhuisgroep Twente) and patients enrolled in the DapKid study (Copenhagen, Steno Diabetes Center). We, therefore, adjusted all our analyses for recruitment location. Determining acute changes in volume markers was beyond the scope of these studies. The currently ongoing study “DAPASALT” (ClinicalTrials.gov identifier: NCT03152084) was designed to prospectively assess short-term and long-term changes in several volume-related markers, such as 24-h urinary sodium, blood pressure, and copeptin. 

In conclusion, this study shows that dapagliflozin increased urinary osmolality, urinary glucose, and fractional lithium excretion, and it decreased blood pressure and body weight. These results suggest that dapagliflozin exerts a diuretic effect through both osmotic and natriuretic diuresis in individuals with type 2 diabetes and kidney damage. During prolonged treatment with dapagliflozin, compensatory mechanisms are activated to restore body-fluid homeostasis.

## Figures and Tables

**Figure 1 jcm-08-00779-f001:**
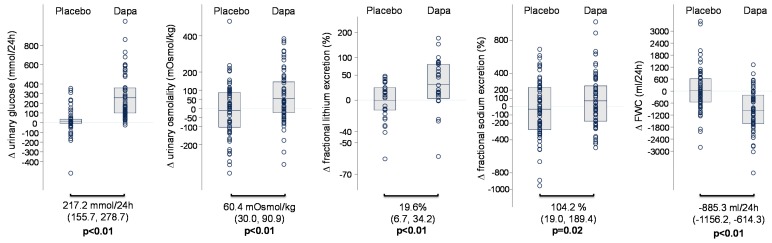
Volume markers at baseline, at the end of placebo treatment, at the end of dapagliflozin treatment, and changes in volume markers during dapagliflozin treatment versus placebo in the merged database (IMPROVE study and DapKid study).

**Figure 2 jcm-08-00779-f002:**
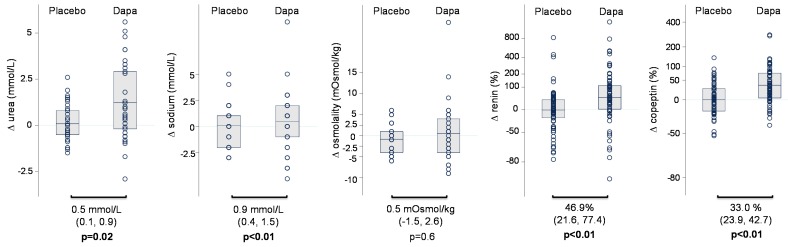
Changes in plasma volume markers during dapagliflozin treatment versus placebo treatment in the merged database.

**Figure 3 jcm-08-00779-f003:**
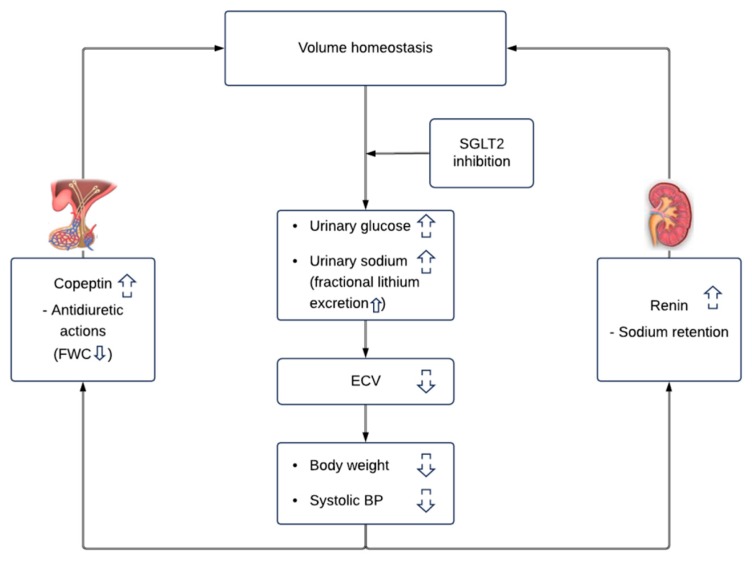
Effects of the sodium glucose co-transporter 2 (SGLT2) inhibitor dapagliflozin on volume homeostasis. Dapagliflozin increases urinary glucose and sodium excretion leading to increased urinary osmolality. Both natriuretic and osmotic diuresis will decrease extracellular volume and decrease body weight and systolic blood pressure. The reduction in extracellular volume activates compensatory mechanisms such as renin and copeptin to restore volume homeostasis. Abbreviations: ECV: extracellular volume, BP: blood pressure, FWC: free water clearance.

**Table 1 jcm-08-00779-t001:** Volume markers at baseline, at the end of placebo treatment, and at the end of dapagliflozin treatment, and changes in volume markers during dapagliflozin treatment versus placebo in the merged database (IMPROVE study and DapKid study).

Characteristics	At Baseline (*n* = 69)	End of Placebo Treatment	End of Dapagliflozin Treatment	Change during Dapagliflozin vs. Placebo (95% CI; *p*-Value)
Weight (kg)	99.2 (21.5)	98.9 (21.2)	97.9 (21.2)	−1.3 (−1.8, 0.9; *p* < 0.01)
Body mass index (kg/m^2^)	31.9 (5.7)	31.8 (5.7)	31.5 (5.8)	−0.39 (−0.6, −0.2; *p* < 0.01)
Systolic blood pressure (mmHg)	141.2 (15.2)	140.4 (14.5)	134.7 (15.9)	−5.7 (−9.1, −2.3; *p* < 0.01)
Diastolic blood pressure (mmHg)	79.8 (8.6)	78.1 (9.4)	76.8 (8.3)	−1.2 (−2.9, 0.5; *p* = 0.2)
Fasting plasma glucose (mmoL/L)	9.8 (3.6)	10.0 (3.4)	8.2 (2.8)	−1.8 (−2.6, −0.9; *p* < 0.01)
HbA_1c_ (mmoL/moL)	65.4 (15.0)	66.6	61.3	−5.2 (−7.2, −3.2; *p* < 0.01)
Sodium (mmoL/L)	139.2 (2.7)	139.6 (2.8)	140.5 (2.8)	0.9 (0.4, 1.5; *p* < 0.01)
Potassium (mmoL/L)	4.3 (0.5)	4.3 (0.4)	4.2 (0.4)	−0.02 (−0.1, 0.1; *p* = 0.61)
Urea (mmoL/L)	6.4 (2.2)	6.6 (2.4)	7.1 (2.6)	0.5 (0.1, 0.9; *p* = 0.02)
Osmolality (mOsmoL/kg)	294.8 (14.4)	291.1 (8.6)	291.6 (7.3)	0.5 (−1.5, 2.6; *p* =0.61)
Copeptin (pmoL/L) ^‡^	8.3 (5.7, 11.2)	8.3 (5.4, 12.6)	11.6 (6.8, 16.6)	33.0% (23.9, 42.7; *p* < 0.01)
Renin (ng/L) ^‡^	37.1 (17.1, 85.0)	33.6 (16.0, 70.1)	59.3 (21.1, 101.0)	46.9% (21.6, 77.4; *p* < 0.01)
NT-proBNP (ng/L) ^‡^	103.0 (35.0, 205.5)	107.5 (43.8, 227.0)	105.0 (48.0, 185)	−5.2% (−19.6, 8.1; *p* = 0.4)
Estimated GFR (mL/min/1.73 m^2^)	79.4 (19.3)	80.1 (18.8)	76.1 (20.8)	−4.1 (−5.9, −2.4; *p* < 0.01)
UACR (mg/g) ^‡^	199.7 (102.3, 405.3)	202.3 (106.3, 480.0)	133.7 (75.3, 282.3)	−52.0% (−72.3, −34.0; *p* < 0.01)
Urinary volume (mL/24 h)	2057 (762)	2120 (741)	2394 (804)	266.3 (100.6, 432.0; *p* < 0.01)
Urine glucose excretion (mmoL/24 h) ^‡^	21.5 (2.0, 130.2)	23.0 (2.0, 154.0)	211.3 (121.1, 512.5)	217.2 (155.7, 278.7; *p* < 0.01)
Urinary osmolality (mOsmoL/kg)	560.7 (177.3)	553.4 (175.6)	614.2 (131.7)	60.4 (30.0, 90.9; *p* < 0.01)
Urinary sodium excretion (mmoL/24 h)	205.2 (110.6)	200.5 (84.5)	195.9 (98.3)	−4.5 (−27.5, 18.5; *p* = 0.70)
Fractional sodium excretion (%)	937.8 (321.2)	898.9 (335.1)	1006.3 (384.8)	104.2% (19.0, 189.4; *p* = 0.02)
Fractional lithium excretion (%)^‡#^	11,318.7 (8984.9, 17,344.4)	10,484.6 (8648.9, 13,734.9)	12,437.4 (10,461.9, 16,275.4)	19.6% (6.7, 34.2; *p* < 0.01)
Free water clearance (FWC) (mL/24 h)	−1727.1 (−1335.4)	−1724.3 (−1230.6)	−2606.1 (−1390.7)	−885.3 (−1156.2, −614.3; *p* < 0.01)

Data are given as mean (SD) and ^‡^ median (25th–75th percentile). # Fractional lithium excretion was only measured in the IMPROVE study and not in the DapKid study.

**Table 2 jcm-08-00779-t002:** Changes in urinary osmolality, NT-proBNP, and copeptin during dapagliflozin therapy versus placebo in various subgroups.

Baseline Subgroups	Mean Baseline Urinary Osmolality (mOsmoL/kg) (SD)	Change in Urinary Osmolality (mOsmoL/kg) (95% CI; *p*-Value)	Median Baseline NT-proBNP (ng/L) (IQR)	Change in NT-proBNP (%) (95% CI; *p*-Value)	Median Baseline Copeptin (pmoL/L) (IQR)	Change in Copeptin (%) (95% CI; *p*-Value)	Median Baseline Renin (ng/L) (IQR)	Change in Renin (%) (95% CI; *p*-Value)
Diuretics use Yes (N = 44) No (N = 25)	554 (168) 573 (196)	63.1 (24.7, 101.5; *p* < 0.01) 59.1 (8.2, 110.0; *p* = 0.02)	75 (35, 201) 113 (38, 206)	−3.8% (−22.2, 13.5; *p* = 0.65) 7.1% (−33.6, 16.4; *p* = 0.54)	7.3 (5.4, 11.2) 9.0 (5.9, 10.9)	29.6% (18.7, 41.6; *p* < 0.01) 39.1% (23.9, 56.1; *p* < 0.01)	38 (19, 85) 29 (17, 72)	52.6% (20.4, 93.4; *p* < 0.01) 38.7% (1.6, 89.3; *p* = 0.04)
p for interaction		*p = 0.19*		*p = 0.39*		*p = 0.41*		*p = 0.08*
HbA_1c_ (mmoL/moL) <63 ≥63	532 (175) 589 (177)	71.7 (27.7, 115.8; *p* < 0.01) 51.4 (9.0, 93.8; *p* = 0.02)	110 (35, 209) 100 (35, 201)	4.1% (-16.2, 25.8; p=0.68) −13.0% (−35.0, 5.8; *p* = 0.18)	7.5 (5.6, 11.2) 9.0 (5.8, 14.0)	32.0% (19.0, 46.5; *p* < 0.01) 35.3% (19.0, 53.7; *p* < 0.01)	29 (19, 87) 40 (15, 85)	32.2% (1.07, 73.0; *p* = 0.04) 62.7% (25.3, 111.2; *p* < 0.01)
p for interaction		*p = 0.51*		*p = 0.76*		*p = 0.49*		*p = 0.56*
Estimated GFR (mL/min/1.73 m^2^) <82 ≥82	509 (152) 611 (188)	65.4 (21.2, 109.5; *p* < 0.01) 57.1 (14.6, 99.6; *p* < 0.01)	118 (53, 263) 66 (35, 150)	−4.8% (−26.1, 14.7; *p* = 0.61) −4.6% (−26.2, 15.3; *p* = 0.63)	9.2 (6.3, 16.4) 7.1 (4.5, 10.0)	36.7% (23.6, 51.2; *p* < 0.01) 29.5% (17.4, 42.8; *p* < 0.01)	29 (20, 113) 41 (11, 70)	48.5% (13.3, 94.7; *p* < 0.01) 46.0% (12.2, 90.0; *p* = 0.01)
p for interaction		*p = 0.68*		*p = 0.85*		*p = b0.46*		*p = 0.66*
UACR (mg/g) <199.7 ≥199.7	591 (199) 532 (151)	76.8 (33.6, 120.0; *p* < 0.01) 46.6 (3.7, 89.5; *p* = 0.03)	75 (35, 201) 112 (53, 263)	−10.3% −33.3, 9.5; *p* = 0.30) −0.1% (−19.8, 20.0; *p* = 0.99)	7.7 (5.8, 10.9) 8.9 (5.7, 11.2)	32.5% (19.9, 46.4; *p* < 0.01) 33.5% (20.8, 47.5; *p* < 0.01)	38 (17, 67) 37 (19, 85)	45.4% (11.3, 89.8; *p* < 0.01) 49.0% (14.1, 94.7, *p* < 0.01)
p for interaction		*p = 0.92*		*p = 0.71*		*p = 0.29*		*p = 0.41*

HbA1c, eGFR, and UACR were split by using the median value (merged database).

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
