# Peer review of "Effects of Dapagliflozin on Volume Status When Added to Renin–Angiotensin System Inhibitors"

_jcm, 2019, doi:10.3390/jcm8060779_

Reviewer 1 Report

This is an interesting study in which Eickhoff and colleague examine the effect of dapagliflozin on volume markers in T2DM patients using data from previously completed clinical studies. The study demonstrates reduction in free water clearance accompanied with increase in plasma renin and copeptin concentrations in dapagliflozin-treated subjects compared to placebo, and increase in lithium fractional extraction. The authors interpreted these results as a compensatory response to decrease in plasma volume produced by dapagliflozin.

Although in general, the results of the study are consistent with previous studies, it conclusions highly relies on the results of other studies.

A major limitation of the study is a lack of direct measurement of plasma/extracellular fluid based on which the authors attributed their observations. A second major limitation is the lack of early measurement (days to one week) of the volume markers, since time dependent changes in body fluids following SGLT2 inhibitors previously has been reported. The authors have to acknowledge these two major limitations in the discussion

How was the increase in plasma renin activity influenced by diuretic use, HbA1c etc (Table 2)? Further, since the increase in plasma renin is a consistent finding, one would anticipate that the magnitude of increase in plasma renin may be affected by ACEI. Thus, it is worth stratifying patients according to baseline ACEI therapy (Table 2)          

Author Response

We uploaded the response to the reviewer's comments as a Word file.

Reviewer 2 Report

The authors performed a post-hoc analysis of two randomized controlled trials (N=69; IMPROVE  and DapKid), assessing the osmotic/diuretic effects of dapagliflozin 10 mg/day when added to renin-angiotensin-system inhibition in patients with type 2 diabetes and urinary albumin-to-creatinine ratio ≥ 30 mg/g. The authors conclude that the observed changes in markers of volume status suggest that dapagliflozin exerts both osmotic and natriuretic effects in patients with type 2 diabetes and kidney damage, as reflected by increased urinary osmolality and fractional lithium excretion. The authors further suggest that compensatory mechanisms are activated to retain sodium and water.

Limitations of the study:

although the study provides further insight into the possible mechanism of action of SLGT2 inhibitors, the data are limited to a post hoc analysis and do not necessarily reflect the effect of the entire class;

no new data are presented, since it is a post-hoc analysis of a limited number of patients.

Strengths:

methods are adequate;

the manuscript is clearly organized and written.

Suggestions for improvement

the authors should add a few paragraphs clearly stating what this study adds to the present state of knowledge;

a diagram illustrating the compensatory mechanisms hypothesized to interpret the findings should be provided;

the criteria used to “merge the databases” of the two trials should be clarified;

the reason why copeptin (a non specific biomarker) was used should be clarified.

Minor point

I noticed a few typos and repetitions in the numbering of the references.

Author Response

The response to the reviewer's comments is uploaded as a Word file.

Reviewer 3 Report

This is a very relevant and interesting study on the "Effects of Dapagliflozin on Volume Status when 3 added to Renin-Angiotensin-System inhibitors"

The study is well conducted and written even though, there is not much novelty. It can be improved by elaborating further in the introduction the need to combine the dapagliflozin with RAAS inhibition. A description of the complimentary modes of action in renal protection through the tubulo-glumerular feedback  process is needed to clarify this process better for readers.

Author Response

Response to Reviewer 3

Comment 1:

The novelty of the manuscript can be improved by elaborating further in the introduction the need to combine the dapagliflozin with RAAS inhibition. A description of the complimentary modes of action in renal protection through the tubulo-glumerular feedback  process is needed to clarify this process better for readers.

Response:

We thank the reviewer for the comment and expanded the introduction based on this comment. The hypothesis of tubuloglomerular feedback has been reviewed in detail in various articles and reviews about SGLT2 inhibitors and is beyond the scope of this study. We therefore did not describe this in the current manuscript.